# Perspectives of Brazilian Primary Care Nurses on Mental Health Care for Hypertensive Older Adults: A Qualitative Study

**DOI:** 10.3390/ijerph20126185

**Published:** 2023-06-20

**Authors:** Clesyane Alves Figueiredo, Daniella Pires Nunes, Suzimar de Fátima Benato Fusco, Maria Giovana Borges Saidel

**Affiliations:** Escola de Enfermagem, Universidade Estadual de Campinas, Campinas 13083-887, Brazil; dpnunes@unicamp.br (D.P.N.); sbenato@unicamp.br (S.d.F.B.F.); mgsaidel@unicamp.br (M.G.B.S.)

**Keywords:** aged, hypertension, mental health, primary health care

## Abstract

Population longevity has been growing globally. In developing countries such as Brazil, the impact of this reality is enormous. The aging process is challenging for the healthcare system, making individuals more susceptible to chronic health conditions and mental health-related diseases. Primary healthcare (PHC) providers must be able to accompany older adults with their singularities in their work processes. This study aims to understand PHC nurses’ perspectives on the mental health care of hypertensive older adults. This is a study with a qualitative approach, using in-depth interviews and a focus group with 16 nurses from the five Brazilian municipalities with the highest number of older adults. The themes that emerged from the data collection were possibilities of PHC, characterization of PHC, and mental health care in PHC. The study findings contribute to the knowledge base on how PHC nurses provide care to hypertensive older adults and which weaknesses they should overcome in their work environment. The different ways providers have been developing strategies to improve their care should be encouraged, improved, and systematized.

## 1. Introduction

Globally, the population’s longevity has been growing, and the projection is that this number will increase progressively in the coming years [1,2,3]. However, aging impairs motor, sensory, and cognitive functions, with greater compromise in persons with chronic conditions [3].

Concerning chronic health conditions, Systemic Arterial Hypertension (SAH) is one of the most prevalent diseases in the population and one of the main risk factors for the development of cardiovascular complications [4]. The prevalence of SAH increases with age, and cardiovascular diseases are the main cause of death in older adults over 65 years old [1].

Cardiovascular diseases (CVD) are associated with depressive disorders and poor long-term outcomes—the literature shows that stroke-related morbidity and mortality can be significantly increased by depression [5]. Chronic health problems, such as SAH, can also precipitate or exacerbate other psychic symptoms, thus causing a greater risk of developing common mental disorders (CMD), pathologies that cause constant physical and emotional stress [6].

CMD is characterized by irritability, insomnia, anxiety, memory loss, and somatic complaints. They are among the most prevalent mental health-related morbidities. Due to the aging process, older adults have a loss of independence and autonomy, which can cause or exacerbate sadness, social isolation, and psychic suffering, conditions that can trigger CMD [6,7]. In addition, studies derived from the Framingham Heart Study already demonstrate an association between loneliness and an increased risk of dementia and cognitive decline [8].

Although not as serious as other mental disorders, CMD is considered an important public health problem due to its high prevalence, high demand for health services, and serious impacts on the health and well-being of the individual [6]. Furthermore, the restrictions imposed by the therapeutic proposals for chronic diseases and the quality of life—which worsens with stress—trigger unpleasant sensations. These sensations directly influence treatment adherence, negatively interfering with the desire to live well and better [9].

Aspects such as those mentioned above, combined with the prerogative that the particularities of senescence favor the aggravation or trigger the presence of mental disorders—such as depression, anxiety, and delusions, among others—reinforce the increased susceptibility of older adults to mental health-related conditions. Approximately 20% of the world’s aged population has a mental health disorder [10].

The greater fragility of older adults implies the need to develop public policies that favor reaching old age with more health and well-being. This is the case of primary care policies, health care for older adults, and humanization in the Brazilian Unified Health System (SUS) [11].

Effective and regular follow-up in primary healthcare (PHC) services can prevent complications and provide a better quality of life, even at an advanced age. The model of public health care in Brazil is offered through the SUS system, and its main gateway is the PHC since PHC facilities are community health services accessible to all citizens and a favorable environment for creating a bond between healthcare providers and patients [12].

PHC professionals are constant caretakers of older adults with chronic problems. Therefore, they must be able to identify the multiple factors that can contribute to the emergence or worsening of mental health problems to set goals and promote and manage the most appropriate interventions. The nurse is a strategic professional in global health, especially in PHC, as nursing care can directly influence health outcomes, reducing morbidity, mortality, and costs and improving people’s quality of life. Therefore, the perception of these professionals is fundamental to developing healthcare strategies [13].

Given the context described above, there is a need to investigate nurses’ perceptions of the mental healthcare of hypertensive older adults to support and implement better strategies for individuals who face this scenario. Therefore, the following research question was elaborated: What structural elements guide the perceptions of primary care nurses about the mental health of hypertensive older adults? Therefore, this study aims to understand PHC nurses’ perspectives on the mental health care of hypertensive older adults.

## 2. Materials and Methods

A qualitative cross-sectional study was conducted using in-depth interviews and a focus group to collect data to identify the perceptions of Brazilian PHC nurses about mental health care for hypertensive older adults.

### 2.1. Study Location

The largest portion of the Brazilian aged population is concentrated in the southeast and south regions of the country, and the five Brazilian cities with the largest number of older adults are Santos, Niterói, Pelotas, Porto Alegre, and Rio de Janeiro [14]. The study was conducted with PHC nurses in the five municipalities above. The Brazilian health system, whose institutional framework was outlined by the 1988 constitution, is still consolidated. The Family Health Strategy (FHS) is the model used to structure primary care in the Unified Health System (SUS). This program organizes primary care facilities focusing on families and communities and integrates health promotion and disease prevention actions [15].

### 2.2. Sample

The sample was intentional and non-probabilistic, with the recruitment of subjects through snowball sampling [16] and sample closing by theoretical saturation [17]. The sample consisted of 16 nurses working in the cities with the largest number of older adults in Brazil, considering the inclusion and exclusion criteria shown in Table 1:

### 2.3. Data Collection

A pilot interview was carried out to adjust the data collection instrument. The pilot testing was conducted with a nurse with seven years of experience and who currently works in a PHC facility in a large city in the countryside of São Paulo, which has many older adults (11.5%, above the Brazilian average) [18]. After conducting the interview, two independent researchers initially performed an audition and analysis to improve the guiding questions. Minor adjustments were made to the vocabulary of the questions to make the questionnaire more accessible to participants. This interview was not considered part of the final sample. It was used only to refine the data collection instrument and facilitate assimilation by the researchers.

The data collection process involved the main researcher sending invitation letters via email to potential participants provided by key informants in the network. The email was sent from one sender to one recipient only. Participants who accepted the invitation were then directed to a virtual environment where they could access the Informed Consent Form (ICF) through Google Forms^®^. After providing consent, a semi-structured interview was scheduled. Five participants did not respond to the email, and none declined consent via the ICF. The main researcher conducted the interviews using four guiding questions:Can you tell me about caring for older adults with hypertension who develop mental health problems?Could you describe your current perception of your work?In your opinion, what improvements could be made to enhance the care of older adults with hypertension and mental health problems in primary healthcare?Could you share your thoughts on the role of other nursing team members in caring for older adults with hypertension and mental health issues?

Data were collected using Google Meet^®^ for one-on-one interviews between the main researcher and a participant. Google Meet^®^ is a video communication service that allows users to record the meeting, a useful feature for reviewing the content or sharing with others. The interviews were conducted from August to October 2022, and each interview lasted 38.5 min, on average, with a total time of 7 h and 36 min. 

Following the one-on-one interviews, a focus group was conducted using Google Meet^®^ and a technique that encourages group discussion on a specific topic to collectively construct research results [19]. The aim of the focus group was to validate the findings of the semi-structured interviews. While all research participants were invited to attend, only five were available, coincidentally one from each city. The main researcher facilitated the 2 h and 14 min meeting and presented the recording units identified in the interviews to prompt discussion and validate the results. The group discussed each topic freely, indicating whether they agreed or disagreed with the units, thus validating the overall theme. The identified recording units included the following:


*“I had contact with mental health in college, but only for a period. If you stop to think about it, a matter of one period, and considering the amount of mental health-related content, it is not enough.”*



*“I stayed one month at the CAPS internship. It went way too fast. I think we have much room to grow concerning mental health.”*



*“The demand is consuming us, and we don’t realize it. We serve, serve, serve those who seek us.”*



*“We can’t search for those who don’t show up [...] Who don’t come to the facility because the demand consumes us.”*


All data and information collected in this research were stored on a local electronic device and will be retained for five years from completion, secured with a password. The responsible researcher deleted all records from any virtual platform, shared environment, or cloud immediately after collecting the data. After five years, the interviews will be deleted from all devices where they were stored.

### 2.4. Data Analysis

After the interviews and the focus group, the transcripts were performed, and the data were analyzed using the Thematic Content Analysis technique [19], supported by the NVivo 1.3 Release Software. 

Thematic content analysis is a method of systematically organizing and identifying patterns of meaning or themes in a dataset. It helps distinguish shared elements in a given topic, whether expressed orally or in writing. However, just because something is common does not necessarily mean it is significant. The patterns identified by the researcher through thematic analysis must be relevant to the study’s focus and research question. In other words, they should help answer the research question and contribute to the study’s purpose [20].

### 2.5. Ethical Procedures

The study was approved by the Institutional Review Board of the State University of Campinas/UNICAMP under protocol number 5.179.564. All participants signed the informed consent documents prior to participation. All current norms for interviews in a virtual environment were respected [21]. After data collection, all data were stored in a data repository at the State University of Campinas [22].

### 2.6. Rigor and Credibility

The sampling considered participants with expertise in the subject, having worked for two years in direct care activities with older adults in PHC [23] in one of the five Brazilian municipalities with the highest number of older adults in the country, to ensure rigor and credibility. The researcher is a master’s student with knowledge in qualitative research and a nurse with three years of experience in PHC in a large city with a proportion of older adults superior to the national average. For greater transferability, the interview questions served as a tool to help the researcher encourage participants to recall their experience of assisting hypertensive older adults in PHC, and participants provided detailed descriptions that led to the collection of extensive data. Periodic research meetings were held between the researchers to examine the transcripts, ensuring the quality of the analysis. The text was preliminary coded and categorized in NVivo, independently. The main researcher coded the data, and the study coordinator (last author) performed validation. The inter-rater reliability of the coding process was calculated using the Kappa index [24]. To calculate the Kappa coefficient, a contingency table was created to identify agreements and disagreements between the two researchers for all codes. A Kappa coefficient of 0.93 was obtained, indicating excellent reliability in the coded transcriptions. The codes, sub-themes, and themes were checked and discussed. Participants provided their comments in a focus group to validate the interview data. All steps of the consolidated criteria for reporting qualitative research (COREQ) guidelines were followed [25].

## 3. Results

The sociodemographic characteristics of the 16 participants are shown in Table 2.

Based on the data, themes and subthemes were identified, as shown in Table 3.

1. The theme “possibilities of PHC” emerged from the strategies that, from the perspective of nurses, can improve mental health care for hypertensive older adults in primary healthcare facilities. Among these aspects, the following stood out: expansion of bonds with individuals and families, proximity to the community in territorial care, and the possibility of using non-pharmacological resources for integrative care.

1.1. The “shared care” sub-theme refers to the socialization groups, consisting of meetings scheduled at pre-determined times with or without pre-established themes. They may have different objectives, such as encouraging physical activity and providing areas for play or listening. People from these groups have contact with the health providers outside the physical structure of the facility and socialize with other patients, as quoted:


*“In this group, we realize that even those with a history of suffering (aged women with mental health problems) and the introverted ones manage to smile during the group activities. They socialize even with us (they used to be very reserved), and then they smile and bring up their health issues (TI).”*



*“We have different groups—walking, circular dance, handicraft […] this is how we try to approach hypertensive elderlies with mental health problems. The primary care unit is the only place we can do that. We manage to create a greater bond with them (TG).”*


1.2. The “care in the community” sub-theme portrays strategies that are carried out in the community, which from the perspective of the participants, promotes the bond between the social actors, providing a better quality of life and insertion in the social context, as shown below:


*“There is a gym suitable for them (hypertensive people with mental health problems), but it is linked to the SUS system. There is a real scenario where we talk about re-signification, where they go, participate in activities, and manage to meet other seniors [...]. We even recently had a wedding of two seniors in there (TVI).”*


1.3. The “complementary therapies” sub-theme is about some activities developed with therapeutic intent, widespread in the Brazilian context. Complementary therapies are used for health promotion and disease prevention, aiming to stimulate the bond between the patient and team and integrate the individual into the social context. The participants reported that the complementary therapies add to the strategies that improve healthcare, improving the bond with the patient and welcoming the patient, as highlighted below:


*“I am a qualified aromatherapist, and in my workplace, I am allowed to perform aromatherapy, color therapy, reiki, and music therapy. I have seen how aged patients have benefited from these therapies, so they seek it [...] They (older adults with mental health problems) say: ‘that’s what I needed that day’, so I think this should also be encouraged. The facility has a welcoming environment with an area designed to make patients feel comfortable (GRU).”*


2. The “characterization of PHC” theme elucidates, through the professionals’ understanding, aspects not in line with the Brazilian guidelines of primary healthcare.

2.1. The “biomedical paradigm” sub-theme covers the participants’ experiences of providing care under a model that prioritizes curative, individual, fragmented, and specialized actions to the detriment of health promotion and prevention actions, in addition to not encouraging listening or treating the individual integrally, as shown below:


*“The assistance here is very centered on the symptoms. You don’t think about the context [...]. Some of our patients are grandparents of drug dealers, others are grandparents of girls with unwanted pregnancies [...]. But that is not heard. People are worried about their symptoms. They are very symptomatic (TCO).”*



*“Another very negative point of this issue of being physician-centered is that they (the physicians) demand much referral to specialists (...). Sometimes they come to their appointments and ask for 8 referrals. It is monstrous for us to deal with all of this, even if needed. I think it is very bad (TAN).”*


2.2. The “unscheduled care for acute complaints” sub-theme is about the participants’ perspectives regarding the care for acute complaints in primary care without prior scheduling or an established care plan. The quotes reveal that the work processes have been mischaracterized as they focus on the complaint and the medical conduct, as shown below:


*“My facility is linked to the family health strategy, but it is an emergency room. It is confusing, as it works as an Emergency Care Unit (GRU).”*



*“The spontaneous demand for treating acute complaints within primary care is a cancer (TVI).”*


3. The “mental health care in PHC” theme addresses some perceptions about the challenges encountered in everyday life to provide mental health care to hypertensive older adults in primary care.

3.1. The “invisibility of mental health in primary health care” sub-theme addresses perceptions about how, sometimes, the mental health of hypertensive older adults is placed in the background, either by prioritizing other comorbidities to the detriment of mental health issues or by lacking knowledge for identifying mental health problems, as the quotes below suggest:


*“So, we try. I try to look at the patient as a whole, but sometimes hypertension takes much more than mental health, and we forget the mental health (TAL).”*



*“I confess that mental health is an area that I often cannot approach because I lack knowledge about it. Sometimes, I’m even afraid to approach mental health issues because I can’t handle the situation as I should since I lack knowledge of it (TI).”*


3.2. The “professional qualification” sub-theme covered some perspectives about professional training, either at the undergraduate level or through continuing education, as shown in the quotes below:


*“In college, for example, I had contact with mental health, but it was only for one period (one semester of the year). If you stop to think about it, one period is not enough. I stayed at CAPS (a psychosocial care center) for a month (in an internship). The internship was too short. We have much to grow concerning mental health (TAL).”*



*“I believe that having prior experience in mental health is an advantage for those providing care to hypertensive older adults since they are likely to develop mental health issues. This topic is barely touched on during the undergraduate program. I have missed it [...]. I don’t think we need a mental health specialization program, but we should be trained to approach the theme and broaden our view about it since it is common to find patients with such issues (TL).”*


3.3. The “service in networks” sub-theme covers some experiences with primary healthcare networks. The participants shared their experiences about providing shared care, developing matrix support in mental health, and participating in case discussions, in addition to referrals for shared care with other services:


*“Some hypertensive older adults develop serious mental health problems. If the physician thinks the problem is out of control, the patient is referred to Psychosocial Care Centers (CAPS). These CAPS we have here communicate very poorly with primary care. We don’t have a strong bond. It’s one patient pushing the other (TG).”*



*“Greater support from the staff working at CAPS would help improve healthcare for hypertensive older adults with mental health problems or hiring psychologists to help us conduct conversations with them. There should be a professional who could dedicate his time exclusively for that (TK).”*


## 4. Discussion

From the results, three major themes were identified: possibilities of PHC, characterization of PHC, and mental health care in PHC, addressing nurses’ perspectives on strategies that facilitate and promote care and how PHC is configured in their workplace.

Within the PHC possibilities, shared care proved to be, from the participants’ perception, a suitable strategy to promote socialization between patients and providers. The group activities mentioned by the participants are diversified and encourage physical activity and listening. Patients can experience an environment of exchange and collective construction of feelings and experiences. Providing an environment for social exchanges with other people who live with the same conditions, either because of their advanced age or health problem, reduces anxiety. It enables psychosocial support, becoming an important health promotion strategy [26]. A study published by the Lancet confirms that community-based services are the best way to extend mental health care to the population [27].

In the “care in the community” sub-theme, the professionals’ perspectives highlighted the importance of being close to the patient and using all possible strategies to build a strong bond within the community. Many participants reported being affected by physical mobility and financial restrictions hampering the trip to the primary healthcare facility. Building a bond with these people favors the therapeutic effect of the groups, even if this is not the main objective, in addition to strengthening relationships and favoring the development of a mutual support network that promotes commitment and co-responsibility [28]. Such a strategy assertively meets the prerogatives of the World Health Organization (WHO), which aims to integrate mental health care into PHC because of the greater ease and proximity of access, reducing healthcare-related expenditures [29].

In the “complementary therapies” sub-theme, the participants’ perspectives highlighted the wide spread of these practices in the Brazilian context, including music therapy, acupuncture, and herbal medicines, among other complementary and integrative therapies. These therapies provide an environment with other alternatives to allopathy and medicalization, enabling health promotion and disease prevention. According to the participants’ perspective, complementary therapies benefit the mental health of hypertensive older adults. Furthermore, it is known that pharmacological agents can have a series of side effects, in addition to generating high expenditure of financial resources, leading to low adherence and, consequently, discontinuation [30]. Today, complementary therapies have high-quality evidence regarding their effectiveness, in addition to spending low allocation of financial resources and side effects that are quite mitigated concerning conventional medications, as demonstrated by a North American systematic review [31]. Therefore, complementary therapies may represent an important step forward in improving the quality of PHC in the Brazilian public healthcare system, adding and even enhancing traditional treatments.

As for the characterization of PHC, the biomedical paradigm sub-theme revealed that the current model of care still goes against the prerogatives of PHC, in which the concepts of multidisciplinarity and comprehensiveness are highlighted. The participants shared some situations that happen in their daily lives and are anchored in the paradigm of the biomedical care model, centered on the symptom and with a high demand for specialties. It is known that the training of health professionals in the Brazilian context is still moving towards a broader understanding of the health–disease process, especially concerning mental health. While some healthcare providers are equipped to adopt a new service model, there is still a predominant focus on traditional theoretical and methodological approaches, typically centered around the traditional psychiatry model [32]. The work processes are marked by the different experiences that each professional has in the field of mental health. Therefore, one of the WHO guidelines says that integrating mental health care and primary care requires the development of training programs for human resources and encouraging comprehensive health care [29].

The sub-theme of “unscheduled care for acute complaints “gathered perspectives on how the providers see their own work process, especially in the reception sector of the PHC facility. They emphasized an overload of professionals for treating acute complaints that could often be directed to other services, such as the Emergency Care Units (Unidades de Pronto Atendimento, UPAs). Furthermore, they reported that this overload of the service results in damage to the programmed and follow-up actions that should occur. Although the participants’ perceptions are based on a more integrative care model, the practice and the care model are based on biomedical logic. If necessary, the nurse sees herself/himself identifying the different dimensions of her/his work process and organizing it as an active and autonomous subject so that it is possible to shape aspects that depend on her/him, as it is also known that there are pre-processes that are independent of their organization [33].

The “mental health care in PHC” sub-theme highlighted the invisibility of mental health in PHC and revealed the professionals’ perspectives on identifying mental health problems in their daily lives. The reports demonstrated a lack of resources for identifying and screening mental health disorders. It is necessary to develop an epidemiological systematization regarding mental health problems in PHC, with identification and description of clinical information so care actions can be planned and implemented. However, the professionals’ perspectives are still based on the traditional model of psychiatry, focusing on the fragmented medical-psychiatric model. Therefore, the visibility of the manifestations of psychic suffering in these individuals depends on the attitude of the professionals in recognizing these manifestations as part of their roles [34]. Thus, it is urgent to open up new opportunities for knowledge and deepen knowledge of mental health.

The “professional qualification” sub-theme covered the participants’ perceptions about their own training and knowledge, which, in most cases, has gaps. These findings reflect the existing structure of knowledge in our society regarding mental health care and strategies from other countries to mitigate such effects, as is the case of health literacy. Health literacy is a dynamic process that develops the individual’s ability to use certain skills beyond literacy on a given topic, allowing the individual to assume an active and critical role on a personal, professional, and social level [35].

Finally, the “service in networks” sub-theme clarified the professionals’ perspectives on shared care, multidisciplinary, and shared knowledge. These experiences tend to be, in the view of the participants, very useful to increase the theoretical framework and improve clinical practice. However, due to multiple weaknesses, shared care is not well established in the Brazilian context of PHC.

### 4.1. Strengths and Limitations 

The potentiality of the study is to construct an intentional and snowball sample made up of nurses from Brazilian municipalities with a high number of older adults and extensive clinical experience in PHC. The robustness of the sample concerning the object of study and the implications of the results are strengths that support clinical practice through naturalistic generalization.

As a limitation, we believe that the field and context studied, Brazilian PHC, presents particularities, such as the socioeconomic level of the population, that are different from developed countries. Another limitation to be considered is that the Brazilian Unified Health System has peculiar characteristics that must be considered during the reading of the results and discussion sections.

### 4.2. Implications

The findings in this study highlight the necessity to broaden health policies that concentrate on community-based mental health services as crucial strategies in this context, along with increased investment in human resources.

In addition to highlighting the importance of enhancing nurse training, whether at the undergraduate level or through periodic courses or the implementation of specific protocols, to encompass the unique care provided to these individuals within the knowledge framework of all professionals, several challenges need to be addressed. Within the context of Brazilian PHC, these challenges include the operational mindset of many administrators, the stigma surrounding mental health care, the workload burden faced by nurses directly involved in patient care, and other obstacles that hinder investment in the preparation of professionals operating within this context. By addressing these challenges, it becomes possible to enhance nurses’ knowledge and ensure improved care for hypertensive older adults with mental health issues.

## 5. Conclusions

This study made it possible to map the perceptions of Brazilian PHC nurses about mental health care for hypertensive older adults. The heterogeneity of the sample stood out, demonstrating important strategies for carrying out this type of healthcare, in addition to weaknesses in different spheres.

Structural problems such as inadequate training of professionals and mischaracterization of the care model are evident and raise challenges that must be considered in the short, medium, and long term in compliance with the guidelines already formalized by the WHO. The need for investment in the comprehensive education of these professionals becomes evident, both during their undergraduate studies and after their entry into the workforce, through continuous training courses or professional development opportunities. This ensures the provision of opportunities for learning and furthering knowledge in this field. Additionally, the various ways the professionals have been developing strategies to improve their care should be encouraged and optimized, expanding the technical capacity of the PHC teams as initiatives are being implemented. This is the case of community-based care, with socialization groups, community service, encouragement of the link between professionals and community, and non-pharmacological therapeutic resources combining physical activity encouragement and social inclusion to improve quality of life and, thus, contribute to comprehensive care and healthy aging.

## Figures and Tables

**Table 1 ijerph-20-06185-t001:** Inclusion and Exclusion Criteria.

Inclusion	Exclusion
Being a PHC nurse Working in the Family Health Strategy in one of the five cities chosen for the study Minimum work experience of two years Being in direct assistance activities	Being on work leave during the data collection period

**Table 2 ijerph-20-06185-t002:** Sociodemographic characteristics of the sample.

Sociodemographic Characteristics		
Sex	Women (%)	93.8
	Men (%)	6.3
Marital status	Married (%)	62.5
	Single (%)	31.3
	Stable union (%)	6.3
Religion	Without religion (%)	31.3
	Catholic (%)	37.5
	Spiritist (%)	18.8
	Protestant (%)	6.3
	Umbandista (%)	6.3
PHC experience time (average)	5.9 years	Standard deviation: 3.92
Average age	39.3 years	Standard deviation: 9.03

**Table 3 ijerph-20-06185-t003:** Themes and Subthemes.

Themes	Sub-Themes
1. Possibilities of PHC	1.1. Shared care 1.2. Care in the community 1.3. Complementary therapies
2. Characterization of PHC	2.1. Biomedical paradigm 2.2. Unscheduled care for acute complaints
3. Mental health care in PHC	3.1. Invisibility of mental health in PHC 3.2. Professional qualification 3.3. Service in networks

## Data Availability

All data were stored in a data repository at the State University of Campinas [22].

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
