# Peer review of "Perspectives of Brazilian Primary Care Nurses on Mental Health Care for Hypertensive Older Adults: A Qualitative Study"

_ijerph, 2023, doi:10.3390/ijerph20126185_

Round 1
Reviewer 1 Report
It is a pleasure to read this manuscript. There are a few minor recommendations, as follows:
Perhaps to use the term older adults rather than elderly
The meaning of this is not clear, "The focus group lasted 2 hours and 14 minutes and was held with 5 participants, one representative of each city. (in data collection). " Did you mean one group from one city?
Did you mean NVivo 13 or 1.3?
I'm afraid I do not understand the meaning of "what is common". I think more elaboration is needed, here, " It is a way of distinguishing "what is common" in a given topic, spoken or written. However, "what is common" is not necessarily significant just because it exists"
Who, and how many researchers coded the data?
May I know if there is a need to state "Brazil 2013" in Table 3?
"The main researcher conducted the interviews using four guiding questions" What were these questions?
"After the interviews, a focus group (FG) was held" - Did you mean that the first wave of interviews were held using in-depth individual interviews, before proceeding to focus group interviews? IT is not clearly stated in the ms. Were these done online or f2f? Please state these details in the methods section.
"Moreover, in our research protocol, the space was used to validate the results of the semi-structured interviews." It is not clear to me what is meant by space, and how this "space" allowed for the validation of the results?
The authors stated that they used four units in the recordings to trigger discussion and validate results. I'm not clear how these recordings were presented, and what were the instructions were given to the participants after hearing these triggers? How were the results validated using these triggers?
In the results section, "1.3. The sub-topic", should it be the sub-theme?
One of the sub-themes, "care in the territory" I feel could be rephrased to delineate what the "territory" means.
There are some statements in the discussion section which needs to have citations mentioned. For example, "Besides, it is known that pharmacological agents can have a series of side effects, in addition to generating high expenditure of financial resources, leading to low adherence and, consequently, discontinuation." There are other examples. Kindly read through the discussion section again to reconsider which statements need to be backed by a citation.
The conclusions should ony summarise the study, and not add new information Some of the implications stated in the conclusions section could be moved to the implications section. May I recommend for the authors to make an implications section to clearly state how the study results could be used to improve mental health care for older adults with hypertensive disease?
Thank you
Author Response
Dear Reviewer,
Thank you for allowing us to revise our manuscript entitled "Perspectives of Brazilian primary care nurses on mental health care for hypertensive older adults: a qualitative study". We appreciate the valuable feedback and constructive criticism provided by the reviewers, which has helped us improve our article's quality. In this letter, we would like to address the reviewers' comments point by point and describe the changes we have made to the manuscript. We believe these revisions have substantially improved the clarity and quality of the manuscript.
REVIEWER 1:
#1: Perhaps to use the term older adults rather than elderly
Response: Thank you for the suggestion. Changes were made as requested.
#2: The meaning of this is not clear, "The focus group lasted 2 hours and 14 minutes and was held with 5 participants, one representative of each city. (in data collection). " Did you mean one group from one city?
Response: Thank you for your feedback. We have adjusted the language and made several changes in this paragraph, providing more detail about how the interviews and focus group were held to improve clarity.
#3: Did you mean NVivo 13 or 1.3?
Response: The correct term is NVivo 1.3. We decided to disclose the software version used in the study.
# 4: I'm afraid I do not understand the meaning of "what is common". I think more elaboration is needed, here, " It is a way of distinguishing "what is common" in a given topic, spoken or written. However, "what is common" is not necessarily significant just because it exists"
Response: Thank you for your feedback. We have adjusted the language and made several changes to this paragraph to improve clarity.
#5: Who, and how many researchers coded the data?
Response: The main researcher coded the data, and the study coordinator (last author) validated it. We have made changes to present this information in the text explicitly. Thank you for bringing this to our attention.
#6: May I know if there is a need to state "Brazil 2013" in Table 3?
Response: There is no need to state that in the Table 3 heading. Thank you for your suggestion.
#7: The main researcher conducted the interviews using four guiding questions" What were these questions?
Response: The questions were inserted in the text to improve clarity. Thank you.
#8: After the interviews, a focus group (FG) was held" - Did you mean that the first wave of interviews were held using in-depth individual interviews, before proceeding to focus group interviews? IT is not clearly stated in the ms. Were these done online or f2f? Please state these details in the methods section.
Response: Thank you for your feedback. To improve clarity, we have adjusted the language and made several changes in the Methods section, providing more detail about how the interviews and focus group were held.
#9: Moreover, in our research protocol, the space was used to validate the results of the semi-structured interviews." It is not clear to me what is meant by space, and how this "space" allowed for the validation of the results?
Response: Thank you for the feedback. The term "space" was not a good choice for the context described in this sentence. We have revised the language to make it clearer.
#10: The authors stated that they used four units in the recordings to trigger discussion and validate results. I'm not clear how these recordings were presented, and what were the instructions were given to the participants after hearing these triggers? How were the results validated using these triggers?
Response: The main researcher conducted the interviews using four guiding questions. In the new version of the article, we have described these questions. Besides, several changes were made to make the language more clear.
#11: In the results section, "1.3. The sub-topic", should it be the sub-theme?
Response: Yes, the correct term is sub-theme. Thank you.
#12: One of the sub-themes, "care in the territory" I feel could be rephrased to delineate what the "territory" means.
Response: The word "territory" was not a good choice for the context. Therefore, the sub-theme was renamed to "Care in the community".
#13: There are some statements in the discussion section which needs to have citations mentioned. For example, "Besides, it is known that pharmacological agents can have a series of side effects, in addition to generating high expenditure of financial resources, leading to low adherence and, consequently, discontinuation." There are other examples.
Response: Thanks for the suggestion. Changes were made, and citations were added as requested.
#14: The conclusions should ony summarise the study, and not add new information Some of the implications stated in the conclusions section could be moved to the implications section. May I recommend for the authors to make an implications section to clearly state how the study results could be used to improve mental health care for older adults with hypertensive disease?
Response: Thank you for your feedback. The suggestion has been accepted, an implications section has been added, and the requested changes have been made.
Reviewer 2 Report
I would like to thank the editors and authors for the opportunity to review the article “Perspectives of Brazilian Primary Care Nurses on Mental Health Care for Hypertensive Elderly: A Qualitative Study”
In general terms, I can state that the article presents a good bibliographic base, with recent citations and specific to the area of interest.
I note some methodological issues in the qualitative analysis, design, etc... that should be elaborated with more precision.
The chosen topic is an interesting one of greater importance considering the increase in longevity of the population and the importance of investing in primary health care to ensure that people receive comprehensive care, including mental health care.
However, little is studied about mental health intervention. The authors are to be congratulated for choosing the topic and having the objective of knowing the perception of Primary nurses on mental health care for hypertensive elderly people.
I will now offer my contributions or suggestions for improving the text:
I suggest clarifying some points:
- The approach method of how the participants were approached, for example. phone, mail, email. If there were people who refused to participate or dropped out and why.
- After conducting the pilot interview, two independent researchers initially performed an audition and analysis to improve the guiding questions. It seems to me that it is important to mention what was changed in the questions and that the authors provide the guide for the 4 guiding questions.
- Where the interviews were held and on average how long each interview lasted. When they refer to 7h and 36 minutes, it seems that it is in all interviews.
- If during the interviews, there was someone else present besides the researchers and the participants.
- It is not clear whether the interviews were audio or visual recorded and whether the transcripts were returned to participants for comments and/or corrections.
- In the description of the sample, when referring to the mean age and the PHC experience time, also include the standard deviation.
- The authors state that “The sampling considered participants with expertise in the subject, having worked for two years in direct care activities with elderly people in PHC.” Which author were they based on for this sample selection criterion ( considered participants with expertise in the subject, having worked for two years).
- After data collection, they were stored. Also clarify if the data were destroyed after the study or how long they decided to keep until their destruction and who had access to them and how they were protected.
- I miss the Kappa indicators that numerically reflect the level of consensus on the main themes identified.
In the analysis the data were clearly presented
It is suggested that the implications for research and practice in a specific field be presented (synthesis)
I hope that my contributions will serve to improve this article and the study you propose.
Thank you very much.
Author Response
Dear Reviewer,
Thank you for allowing us to revise our manuscript entitled "Perspectives of Brazilian primary care nurses on mental health care for hypertensive older adults: a qualitative study". We appreciate the valuable feedback and constructive criticism provided by the reviewers, which has helped us improve our article's quality. In this letter, we would like to address the reviewers' comments point by point and describe the changes we have made to the manuscript. We believe these revisions have substantially improved the clarity and quality of the manuscript.
REVIEWER 2:
#1: The approach method of how the participants were approached, for example. phone, mail, email. If there were people who refused to participate or dropped out and why.
Response: Thank you for the suggestion. Changes were made in the text to provide the clarifications needed.
#2: After conducting the pilot interview, two independent researchers initially performed an audition and analysis to improve the guiding questions. It seems to me that it is important to mention what was changed in the questions and that the authors provide the guide for the 4 guiding questions.
Response: Thank you for the suggestion. Changes were made in the text to provide the clarifications needed.
#3: Where the interviews were held and on average how long each interview lasted. When they refer to 7h and 36 minutes, it seems that it is in all interviews.
Response: Thank you for the suggestion. We made several changes in the Methods section to improve clarity.
#4: If during the interviews, there was someone else present besides the researchers and the participants.
Response: Data was collected using Google Meet® for one-on-one interviews between the main researcher and a participant. We have made this information available in the new version of the manuscript.
#5: It is not clear whether the interviews were audio or visual recorded and whether the transcripts were returned to participants for comments and/or corrections.
Response: Google Meet® allows users to record the meeting, a useful feature for reviewing the content or sharing with others. We have made this information available in the new version of the manuscript.
#6: In the description of the sample, when referring to the mean age and the PHC experience time, also include the standard deviation.
Response: Thank you for your feedback. We have inserted the standard deviation values as requested.
#7: The authors state that "The sampling considered participants with expertise in the subject, having worked for two years in direct care activities with elderly people in PHC." Which author were they based on for this sample selection criterion ( considered participants with expertise in the subject, having worked for two years).
Response: We have added a citation to this paragraph, as requested.
#8: After data collection, they were stored. Also clarify if the data were destroyed after the study or how long they decided to keep until their destruction and who had access to them and how they were protected.
Response: All data and information collected in this research were stored on a data repository at the State University of Campinas and will be retained for five years from completion, secured with a password. The responsible researcher deleted all records from any virtual platform, shared environment, or cloud immediately after collecting the data. After five years, the interviews will be deleted from all devices where they were stored. We have included this information in the new version of the article. Thank you!
#9: I miss the Kappa indicators that numerically reflect the level of consensus on the main themes identified.
Response: A Kappa coefficient of 0.93 was obtained, indicating excellent reliability in the coded transcriptions. We have added this information to the new version of the manuscript, as requested.
#10: It is suggested that the implications for research and practice in a specific field be presented (synthesis)
Response: Thank you for your feedback. The suggestion has been accepted, an implications section has been added, and the requested changes have been made.
Round 2
Reviewer 1 Report
Thank you for the opportunity to review this paper again, and I find it very much improved, thank you for considering the suggestions made.
I still find "Brazil, 2023" in Table 2 heading, perhaps it could be deleted if not relevant.
Yes, perhaps it is a good idea to mention the version of the NVivo nevertheless.
For this important article, I would suggest that a stronger implication be made, so that all the resources and efforts invested in this research is worthwhile. What are the implications for the nurses' training, for example, or for other considerations on the care of older adults mentally, based specifically on the study findings?
Thank you.
Author Response
Response letter, in reference to Manuscript ID ijerph-2344499
Type of manuscript: Article
Title: Perspectives of Brazilian primary care nurses on mental health care for hypertensive older adults: a qualitative study
REVIEWER 1:
Thank you for the opportunity to review this paper again, and I find it very much improved, thank you for considering the suggestions made.
Point 1:
I still find "Brazil, 2023" in Table 2 heading, perhaps it could be deleted if not relevant.
Response: We have considered your suggestion, and "Brazil, 2023" has been removed.
Point 2:
Yes, perhaps it is a good idea to mention the version of the NVivo nevertheless.
Response: Thank you.
Point 3:
For this important article, I would suggest that a stronger implication be made, so that all the resources and efforts invested in this research is worthwhile. What are the implications for the nurses' training, for example, or for other considerations on the care of older adults mentally, based specifically on the study findings?
Response: Thank you very much for your valuable suggestion. We have incorporated your feedback by expanding section 4.2 and restructuring section 5 to accommodate the proposed suggestion.
Thank you.